# A putative structural mechanism underlying the antithetic effect of homologous RND1 and RhoD GTPases in mammalian plexin regulation

Yanyan Liu[1], Pu Ke[2], Yi-Chun Kuo[3], Yuxiao Wang[3], Xuewu Zhang[3]*, Chen Song[1,4]*, Yibing Shan[5]*

[1]Center for Quantitative Biology, Academy for Advanced Interdisciplinary Studies, Peking University, Beijing, China; [2]Beijing Computational Science Research Center, Beijing, China; [3]Department of Pharmacology, University of Texas Southwestern Medical Center, Dallas, United States; [4]Peking-Tsinghua Center for Life Sciences, Academy for Advanced Interdisciplinary Studies, Peking University, Beijing, China; [5]Antidote Health Foundation for Cure of Cancer, New York, United States

**Abstract** Plexins are semaphorin receptors that play essential roles in mammalian neuronal axon guidance and in many other important mammalian biological processes. Plexin signaling depends on a semaphorin-induced dimerization mechanism and is modulated by small GTPases of the Rho family, of which RND1 serves as a plexin activator yet its close homolog RhoD an inhibitor. Using molecular dynamics (MD) simulations, we showed that RND1 reinforces the plexin dimerization interface, whereas RhoD destabilizes it due to their differential interaction with the cell membrane. Upon binding plexin at the Rho-GTPase-binding domain (RBD), RND1 and RhoD interact differently with the inner leaflet of the cell membrane and exert opposite effects on the dimerization interface via an allosteric network involving the RBD, RBD linkers, and a buttress segment adjacent to the dimerization interface. The differential membrane interaction is attributed to the fact that, unlike RND1, RhoD features a short C-terminal tail and a positively charged membrane interface.

*For correspondence:
xuewu.zhang@utsouthwestern.
edu (XZ);
c.song@pku.edu.cn (CS);
ybshan@gmail.com (YS)

## Introduction

Plexins are a family of nine single-pass transmembrane receptor proteins including plexin A1–4, B1–3, C1, and D1. Plexins are best known as the receptors of extracellular semaphorin ligands (*Nishide and Kumanogoh, 2018*) that are guidance cues for neuronal axons. Plexins also help regulate other essential biological processes such as cell migration, angiogenesis, and immune responses (*Sakurai et al., 2012*; *Takamatsu and Kumanogoh, 2012*). Aberrant plexin activity is associated with a plethora of diseases including neurological disorders and cancer metastasis (*Gu and Giraudo, 2013*; *Tamagnone, 2012*; *Yaron and Zheng, 2007*).

Plexin architecture is conserved across the family. Plexin consists of a large multi-domain extracellular module including the ligand-binding Sema domain, a single-pass transmembrane helix, and an intracellular module that includes a GTPase-activating protein (GAP) domain and a Rho-family GTPase-binding domain (RBD) (*Figure 1A*). In plexin signaling, semaphorin binds at the extracellular module, which leads to activation of the GAP domain. Structures (*Janssen et al., 2010*; *Kuo et al., 2020*; *Liu et al., 2010*; *Nogi et al., 2010*) showed that a semaphorin mediates plexin dimerization: a semaphorin dimer interacts with two plexins at the extracellular module, and this extracellular dimerization leads to dimerization at the intracellular module (*Figure 1B*), and in turn activation of the GAP domain for the substrate Rap GTPases (*Wang et al., 2012*). The dimerization stabilizes the

**Figure 1.** Plexin architecture, dimerization, and GTPase binding of the RBD domain. (**A**) Components of a plexin molecule. Each RBD domain is connected with a GAP domain by the N and C linkers. (**B**) Architecture of the semaphorin-induced plexin dimer. A buttress segment is positioned between the RBD domain and the dimerization helix. Helix 11 is C-terminal to the buttress. The activation segments are held in the active conformations by the dimerization helices in trans. A Rap GTPase is bound to each GAP domain as a substrate at the active site. (**C**) The RND1- or RhoD-bound plexin dimer systems simulated in this study. The extracellular portions of the dimers were excluded. (**D**, **E**) Structural basis of RND1 stabilization and RhoD destabilization of the plexin dimer according to this study. The key difference is that the catalytic domain of RND1 is relatively detached from the membrane.

The online version of this article includes the following figure supplement(s) for figure 1:

**Figure supplement 1.** RhoD binding with plexin RBD domain and anchoring to the membrane.

active conformation of the so-called activation segments of the GAP domains, which otherwise adopts an inactive conformation that precludes Rap binding to the GAP domain (*Wang et al., 2013*). In plexin signaling, the GAP activity switches off the signaling of plexin substrate Rap by catalyzing its GTP hydrolysis and converting it from the GTP-bound state to the GDP-bound state. We will refer to GTPases such as Rap, which bind at the active site of the GAP domain, substrate GTPases.

Besides the GAP domain, the intracellular module of plexin includes an RBD domain that binds Rho-family GTPases. The RBD of plexin B1 has been shown to bind Rac1, Rac2, Rac3, Rnd1, Rnd2, Rnd3, and RhoD, but not RhoA, Cdc42, RhoG, or Rif (*Fansa et al., 2013*). These Rho-family GTPases to various degrees serve as regulators in plexin activation. To distinguish them from the GAP-binding substrate GTPases such as Rap, we refer to these RBD-binding GTPases as regulatory GTPases.

The Rho-family regulatory GTPases play important roles in plexin regulation from the intracellular environment. Overexpression of Rac1 leads to higher cell surface expression of plexin and enhances plexin interaction with semaphorin, suggesting that Rac1 acts as an upstream activator of plexin (*Vikis et al., 2002*). Binding of overexpressed RND1 to plexin triggers cell collapse in the absence of semaphorin, suggesting that RND1 is a more potent activator than Rac1 for plexin (*Zanata et al., 2002*). Simultaneous extracellular binding of semaphorin and intracellular RBD binding of certain regulatory GTPases appear to be a prerequisite for full activation of at least some plexins

(*Bell et al., 2011*), but RBD binding with some other regulatory GTPases of the Rho family attenuates plexin activity. RhoD and RND1 are two such regulatory GTPases. RhoD binds plexin RBD with similar affinity as RND1 (*Fansa et al., 2013*), but it strongly inhibits plexin signaling (*Zanata et al., 2002*) rather than activates it.

The structural mechanism of the antithetic effects of RhoD and RND1 on plexin signaling, however, remains elusive. Activity assays in solution showed that the Rho-family GTPases do not alter the GAP activity of plexin either in the monomeric or the active dimer state (*Wang et al., 2012*). RND1 is anchored to the membrane by a C-terminal amphipathic helix (*Figure 1C*) and RhoD by lipidation of a cysteine at the C-terminal tail (*Figure 1—figure supplement 1*), and the membrane may play an important role in their plexin regulation. Resolved complex structures of plexin RBD with different Rho-family GTPases, such as RND1 (PDB 2REX and 3Q3J) and Rac1 (*Wang et al., 2012*), showed that the RBD binds with these GTPases in a similar mode. The structure of plexin RBD in complex with RhoD is not available, but data from NMR chemical shift analyses as well as binding assays with RhoD in different nucleotide states and mutants of plexin RBD together suggested that the binding mode is similar (*Fansa et al., 2013*; *Tong et al., 2007*; *Zanata et al., 2002*). The crystal structures and other biophysical data all suggest that the RBD domain does not undergo substantial or global conformational changes upon binding with Rho-family GTPases (*Bell et al., 2011*; *Tong et al., 2007*; *Wang et al., 2012*). Modulations of plexin activity from Rho-family GTPases thus are unlikely to be mediated by major conformational changes within the RBD domain.

To understand the apparent paradox regarding the antithetic effects of RND1 and RhoD on plexin activation, we determined the crystal structure of the RhoD/plexin B2-RBD complex, which confirmed that the RBD binding mode of RhoD is similar to that of other Rho-family regulatory GTPases. We then modeled and simulated plexin A4 complexed with RND1 or RhoD (*Figure 1C*) to investigate the structural mechanisms underlying RND1 as an activator and RhoD as an inhibitor in plexin regulation. The simulations suggested that RND1 binding is compatible with the dimerization of plexin A4 while RhoD binding is likely disruptive to the dimerization. The simulations generated two distinct modes of interactions of RND1 and RhoD with the membrane: RND1 interacts with the membrane loosely and its long C-terminal tail serves as a flexible tether to the membrane (*Figure 1D*), whereas RhoD interacts with the membrane in a specific manner using a positively charged membrane interface (*Figure 1E*), which is absent in RND1. As a result, RND1 binding strengthens plexin dimerization by stabilizing the RBD position with respect to the GAP domain and in turn stabilizing the adjacent dimerization interfaces, while RhoD distorts the RBD position and hinders plexin dimerization.

## Results

### Crystal structure of the complex between RhoD and the plexin B2-RBD domain

Complex structures of the RBD domain with plexin activators such as RND1 or Rac1 have been previously resolved. To experimentally determine the binding mode between plexin and RhoD, a negative plexin regulator, we screened various combinations of RhoD and the intracellular region of plexin family members from different species for crystallization, which resulted in crystals of the complex of mouse plexin B2 and human RhoD bound to the GTP analogue GMP-PNP. Analyses of the diffraction data suggested that plexin B2 degraded during the incubation in crystallization drops, and the crystals only contained the complex between RhoD and the RBD of plexin B2. We solved the structure to 3.1 Å resolution by molecular replacement (*Supplementary file 1*; see Materials and methods for details). The asymmetric unit of the crystal contains two RhoD molecules, each of which binds to one plexin B2 RBD molecule. Surprisingly, the two RBD domains form a domain-swapped dimer in the structure, with the N-terminal portion of one molecule fold together with the C-terminal portion of the other (*Figure 2—figure supplement 1*). This domain-swapped dimer is likely a crystallization artifact because it cannot form in the context of a plexin dimer (*Wang et al., 2013*), in which the two RBD domains are far apart from one another (*Figure 1B*). We therefore consider each RBD domain formed by the two halves of the two molecules as a representative of one intact, unswapped RBD as its conformation is very similar to the structures of other RBDs (*Figure 2A, B*).

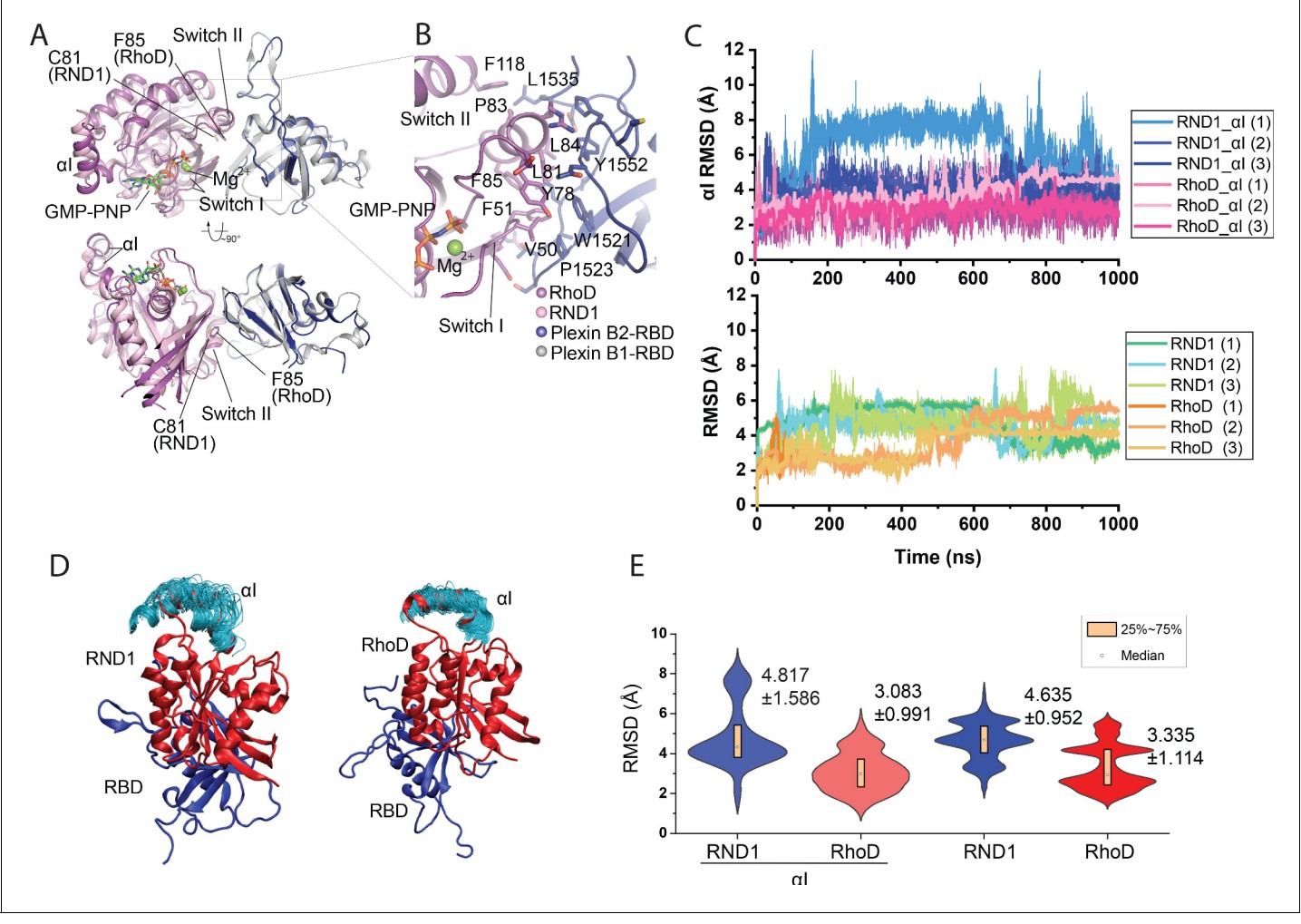

**Figure 2.** Crystal structure of the RhoD/plexin B2-RBD complex. (**A**) Overall structure of the RhoD/plexin B2-RBD complex based on the domain-swapped dimeric structure (*Figure 1—figure supplement 1*). The structure of RND1/plexin B1-RBD complex (PDB ID: 2REX) is superimposed based on the RBD for comparison. (**B**) Expanded view of the binding interface between RhoD and plexin B2-RBD. (**C**) The Cα root mean square deviation (RMSD) of the RND1 and RhoD catalytic domains and their respective αI helices with respect to the initial positions in simulations of the RND1-RBD and RhoD-RBD complex structures (three 1-μs-long simulations for each system). In calculating the RMSDs, the RBD domains were aligned. (**D**) Multiple snapshots of the αI helix in the simulations (with the RBD aligned). As shown, the αI of RND1 appears more flexible in the simulations. (**E**) The RMSD data shown in (**C**) represented in violin plots; the average and RMSD of the distributions are labeled. Consistent with the visualization shown in (**D**), RND1 appears to be more flexible conformationally when bound with RBD.

The online version of this article includes the following figure supplement(s) for figure 2:

**Figure supplement 1.** Two orthogonal views of the asymmetric unit of the RhoD/plexin B2-RBD complex crystal.

The structure confirms that RhoD binds the plexin B2 RBD in a mode similar to those of other complexes between Rho-family GTPases and plexin (*Bell et al., 2011*; *Wang et al., 2011*; *Wang et al., 2012*). The GTP analogue GMP-PNP and $Mg^{2+}$ together stabilize the ligand-binding switch I and switch II regions in the active conformation, which make an extensive interface with one side of the beta-sheet of the RBD (*Figure 2A, B*). All the residues in RhoD involved in interacting with the RBD are identical between human and mouse RhoD, suggesting that the cross-species complex that we crystallized is a valid representative of the RhoD/plexin complex. Interestingly, a superimposition of the RhoD/plexin B2-RBD complex with the RND1/plexin B1-RBD complex based on the RBD domains shows that the orientation of RhoD and RND1 relative to the RBD domains is slightly different (*Figure 2A, B*). Compared with that in RND1, the switch II helix in RhoD is placed further away from the RBD. This appears to be required to accommodate Phe85, which is bulkier

than Cys81, the corresponding RND1 residue. This difference leads to different pivots of the two GTPases relative to the RBD, which propagates to a larger difference in the opposite side of the molecule, where the insert helices (αI), a helical segment uniquely present in the catalytic domains of Rho-family GTPases, is located (*Figure 2A*). In the context of the active dimer of full-length plexin on the plasma membrane, the αI helix faces the membrane (*Figure 1D, E*). This orientational difference of RhoD and RND1 relative to plexin therefore may affect their interactions with the membrane, although it is unclear whether and how that effect is related to the opposite roles of RND1 and RhoD in plexin signaling.

To investigate the conformational dynamics of RBD complexes with RND1 and RhoD, we simulated the RBD complexes with RND1 or RhoD, each for three 1-µs simulations. In these simulations, only the RBD domain of plexin A4 and the catalytic domain of RND1 or RhoD were included. These simulations showed that both complexes are overall stable, with the root mean square deviation (RMSD) of the Cα atoms of the catalytic domains with respect to their initial positions fluctuating around 4 Å when the RBD domain aligned (*Figure 2C*). By this metric, RND1 appears to be more flexible than RhoD relative to the RBD (*Figure 2D*). The αI helix of RND1 was also more flexible than RhoD with respect to the catalytic domain as a whole in the simulations (*Figure 2D, E*). This analysis is consistent with our finding that the RhoD regulation of plexin activity requires a stable membrane interaction involving the αI helix, but the RND1 function involves little membrane interaction. This simulation finding will be discussed in detail later in this report.

## RhoD and RND1 interact differently with cell membrane

The membrane may play an important role in plexin regulation by Rho-family GTPases, which are located adjacent to the membrane. Previous studies showed that RhoD does not alter the GAP activity of plexin A1 in a solvent environment (*Pascoe et al., 2015*; *Wang et al., 2012*). To investigate how the membrane might play a role in plexin regulation mediated by RND1 and RhoD, we simulated plexin A4 dimer in the membrane environment, respectively, bound with RND1 and RhoD (*Figure 1C*). We first constructed a structural model of the transmembrane and the intracellular modules of the plexin A4 dimer with a membrane, primarily using homology modeling based on the resolved structure of the intracellular module of plexin C1 dimer (PDB 4M8N) (*Wang et al., 2013*). We then added two GTP-bound RND1 molecules to the plexin dimer to bind the RBD domains; the RND1 C-terminal tails each forms an amphipathic helical tail (*Thiyagarajan et al., 2004*) and serves as a membrane anchor (*Figure 1C, D*). We similarly constructed a plexin dimer model in which each plexin RBD is bound with a (GTP-bound) RhoD, where Cys207 residue of the C-terminal tail is palmitoylated and anchored to the membrane (*Figure 1E* and *Figure 1—figure supplement 1*). In cells, the membrane anchor of RhoD is more commonly the geranylated (*Hodge and Ridley, 2016*) Cys207, but this difference should not affect the results of the simulations. It is worth noting that the construction of these two models was essentially constrained by existing crystal structures. It involved piecing together the plexin dimer structure (in which the RBD domains are resolved) and the complex structures of RBD bound with RND1 or RhoD. As shown in *Figure 1C*, the positioning of the plexin dimer with respect to the membrane is determined by symmetry, that is, the two halves of the plexin dimer are identical in terms of their positions relative to the membrane. With the exception of the C-terminal loops of RND1 and RhoD, these two models are highly similar prior to simulations.

We simulated the RND1-bound (*Figure 3A*) and the RhoD-bound (*Figure 3B*) plexin dimers, each for 1 µs three times. In the simulations of the RND1-bound dimer, the amphipathic helices at the C-termini of the RND1 molecules remained anchored to the membrane, and the RND1 linkers between the catalytic domains and the amphipathic helices (residues 189–200) are sufficiently long to not affect the position of the catalytic domains (*Figure 3A*). The contact area between the membrane and the catalytic domains remains relatively small, with a mean at approximately 200 Å² (*Figure 3D, E* and *Figure 3—figure supplement 1A*). The two RND1 catalytic domains largely remained in their initial positions, with the RMSD of the Cα atoms with respect to their initial positions fluctuating around 6 Å (*Figure 3E*).

The C-terminal tail of the RhoD is shorter and more arginine-rich than the RND1, which is likely membrane-bound and hence restrains the RhoD catalytic domain to the membrane. In contrast to the RND1-bound plexin dimer, in the simulations of the RhoD-bound plexin dimer, the membrane interactions of the RhoD catalytic domains developed extensive and stable interactions with the

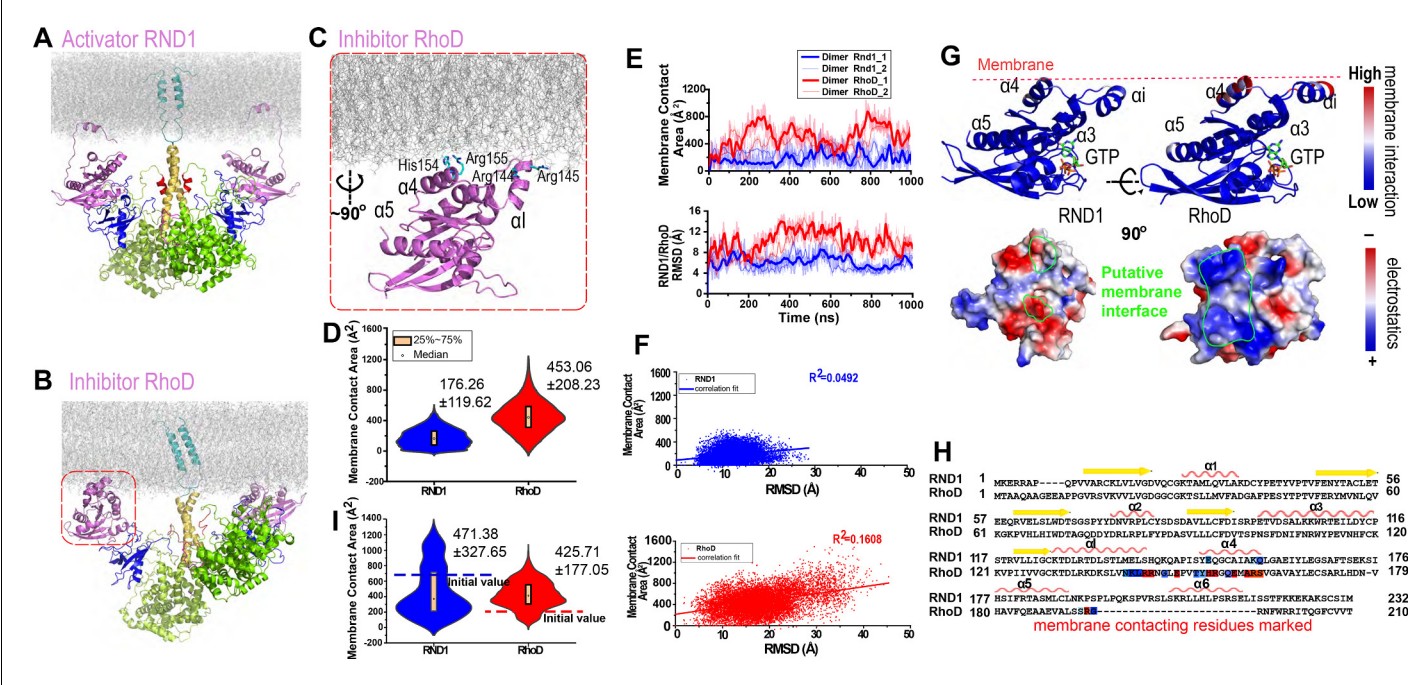

**Figure 3.** Plexin-bound RND1 and RhoD interact with the membrane differently. (**A, B**) Representative snapshots of the simulations of RND1- and RhoD-bound plexin dimer. (**C**) Close-up view of the membrane interaction of RhoD bound with the plexin dimer. Primarily the membrane interface consists of the positive-charged residues of RhoD at the α4 and the αI helices. (**D**) Distributions of the membrane contact area of RND1 and RhoD bound with the plexin dimer. The data was compiled from three simulations each for the RND1- or RhoD-bound dimers. The average and root mean square deviation (RMSD) of each distribution are shown. The occurrence of apparent negative contact area is due to irregularity of the solvent-area program in cases of two objects being adjacent but not in contact with one another. (**E**) The time series of the membrane contact area of RND1 or RhoD in juxtaposition with the time series of the RMSD of the RND1 or RhoD catalytic domains with respect to their initial positions in two representative simulations. (**F**) Scatter plots of the membrane contract area and RMSD data shown in (**E**). As shown, the correlation is stronger for RhoD than for RND1. (**G**) Upper panels: the membrane contact residues of RND1 and RhoD indicated by color coding (the color coding indicates the number of lipid residues within 5 Å of the residue average in all simulations of the RND1- or RhoD-bound plexin dimer); lower panels: the surface electrostatic properties of RND1 and RhoD around their respective membrane-contacting regions. (**H**) Sequence alignment of RND1 against RhoD showing that (1) the positively charged membrane-contacting residues of RhoD are mostly not conserved in RND1, and (2) the C-terminal tail of RhoD is much shorter than that of RND1. The color coding of the membrane contact residues is inherited from (**G**). (**I**) RND1 and RhoD membrane contact areas obtained from the control simulations of plexin dimer, in which RND1 molecules were initiated at positions with large membrane contact while RhoD molecules were initiated at positions with little membrane contact (marked by the dashed lines). Further data from these control simulations are shown in *Figure 3—figure supplement 1B, C*. The average and RMSD for each distribution are shown.

The online version of this article includes the following figure supplement(s) for figure 3:

**Figure supplement 1.** Additional data on RND1/RhoD-membrane contact and charged lipid enrichment.

membrane in the courses of the simulations (*Figure 3D* and *Figure 3—figure supplement 1A*). The contact area of the two RhoD domains with the membrane fluctuated but generally trended upwards. It is apparent that the extent of the membrane interaction is closely correlated with the positioning of the catalytic domains in both the RND1- and the RhoD-bound plexin dimers. With the increase of the membrane interactions, the two RhoD domains deviated substantially from their initial positions, as shown by the RMSD of Cα atoms with respect to their initial positions (*Figure 3E*). The RMSD fluctuation of the RhoD domains was larger than the RND1 domains (*Figure 3E*), indicating that the differential membrane interactions of RhoD and RND1 lead to their differential positioning and dynamics. Our analysis showed that, for RhoD more than for RND1, the membrane contact area is correlated with the RMSD of the GTPase domain with respect to its initial position (*Figure 3F*), in agreement with the notion that the membrane interaction modulates RhoD positioning.

Further analysis suggested that RhoD interacts with the membrane with a specific interface involving the αI helix and the α4 helix (*Figure 3C*); Arg144, Arg145, His154, and Arg155 in this part of

RhoD enjoyed stable interactions with the membrane (*Figure 3G*). The RhoD membrane-anchoring interface features pronounced positive electrostatic potential that is favorable for membrane interaction (*Figure 3G*). As shown in *Figure 3—figure supplement 1D*, the number of RhoD residues in membrane contact grew in the simulations, and the contact map showed that the membrane contact primarily involved the αI helix (residues 130–146) and its neighboring region. We also observed the development of enrichment of the negatively charged 1-palmitoyl-2-oleoyl-sn-glycero-3-phospho-L-serine (POPS) lipids among the lipids in contact with RhoD in the course of the simulations (*Figure 3—figure supplement 1E*), in agreement with the electrostatic nature of the membrane interaction of RhoD. The trend of charged lipids becoming enriched in the membrane interfaces with RhoD molecules was also observed (*Figure 3—figure supplement 1F*) in similar simulations where phosphatidylinositol (4,5)-bisphosphate (PIP2) lipids were included in the membrane. In contrast, RND1 interaction with the membrane is much less stable, without a specific membrane interface (*Figure 3G*) and with fewer residues involved (*Figure 3—figure supplement 1D*). The positively charged residues in the RhoD membrane interface are almost all replaced in RND1 (*Figure 3H*), and hence the strong electrostatic feature of RhoD in that region is absent in RND1 (*Figure 3G*). These observations combined suggest that the tight membrane interaction of RhoD may be attributed to the short C-terminal tail and to the positively charged surface patch, which distinguish RhoD from RND1.

The simulations of both the RND1-bound and the RhoD-bound plexin dimer were initiated from highly similar models that integrate structural information in existing crystal structures. In these simulations, the behavior of RND1 and RhoD molecules diverged in terms of their membrane interaction. To ensure that this finding is not associated with a feature of the particular initial models, we performed the following control simulations. We took a typical simulation-generated conformation of the RhoD-bound plexin dimer (in which the RhoD molecules bear extensive membrane interaction) and swapped the RhoD molecules for RND1. In the resulted RND1-bound plexin dimer, the RND1 molecules inherited the extensive membrane interaction. Similarly, we took a typical simulation-generated conformation of the RND1-bound plexin dimer (in which the RND1 molecules bear limited membrane interaction) and swapped the RND1 molecules for RhoD. In the resulting system, the RhoD molecules bear little membrane interaction. We then performed three 0.5-μs-long simulations for each of these two systems, hoping that the simulations will reinstall the previously observed pattern of membrane interaction for RND1 and RhoD in spite of the initial structures being of the opposite pattern (we added PIP2 lipids to the membrane in these simulations to better represent the cell membrane). Indeed, in the control simulations, the RND1 membrane contact area dwindled, while the RhoD membrane contact area grew steadily and gradually surpassed the RND1 (*Figure 3I* and *Figure 3—figure supplement 1B, C*). We believe that these additional simulations provide important validation to the key observations we obtained with respect to the membrane interactions of RND1 and RhoD. While the trend is clear in these relatively short (0.5-μs-long each) simulations, most likely the simulations have not converged and the trend will become more pronounced if the simulations are extended.

## The differential membrane interactions lead to different RBD position and dynamics

In the plexin dimer, each RhoD or RND1 molecule is located in a space confined by the membrane and the RBD domain and interacts with both simultaneously (*Figure 4A, B*). In the simulations, the RBD interacts with either RhoD or RND1 stably, although RhoD interacts with RBD with a slightly larger interface than RND1 (*Figure 4C* and *Figure 4—figure supplement 1A*). In addition to RND1- and RhoD-bound plexin dimers, we simulated plexin monomer and dimer with the RBD domains unoccupied, each for 500 ns. We analyzed the positions of the RBD domain with respect to the GAP domain in all our simulations. The RBD domain appeared to be inherently flexible with respect to the GAP domain as shown in the monomer simulations (*Figure 4E* and *Figure 4—figure supplement 1C*). This is suggested by existing crystal structures of plexins, in which the RBD domain exhibited substantial flexibility with respect to the GAP domain (*Figure 4F*). The interface between RBD and GAP appeared to be reduced by the presence of RhoD but not by the presence of RND1 (*Figure 4D* and *Figure 4—figure supplement 1B, E*). The RBD RMSD with respect to its initial position was overall larger for the RhoD-bound than for the RND1-bound plexin dimer (*Figure 4E* and *Figure 4—figure supplement 1C*), indicating that RhoD likely displaces the RBD from its native

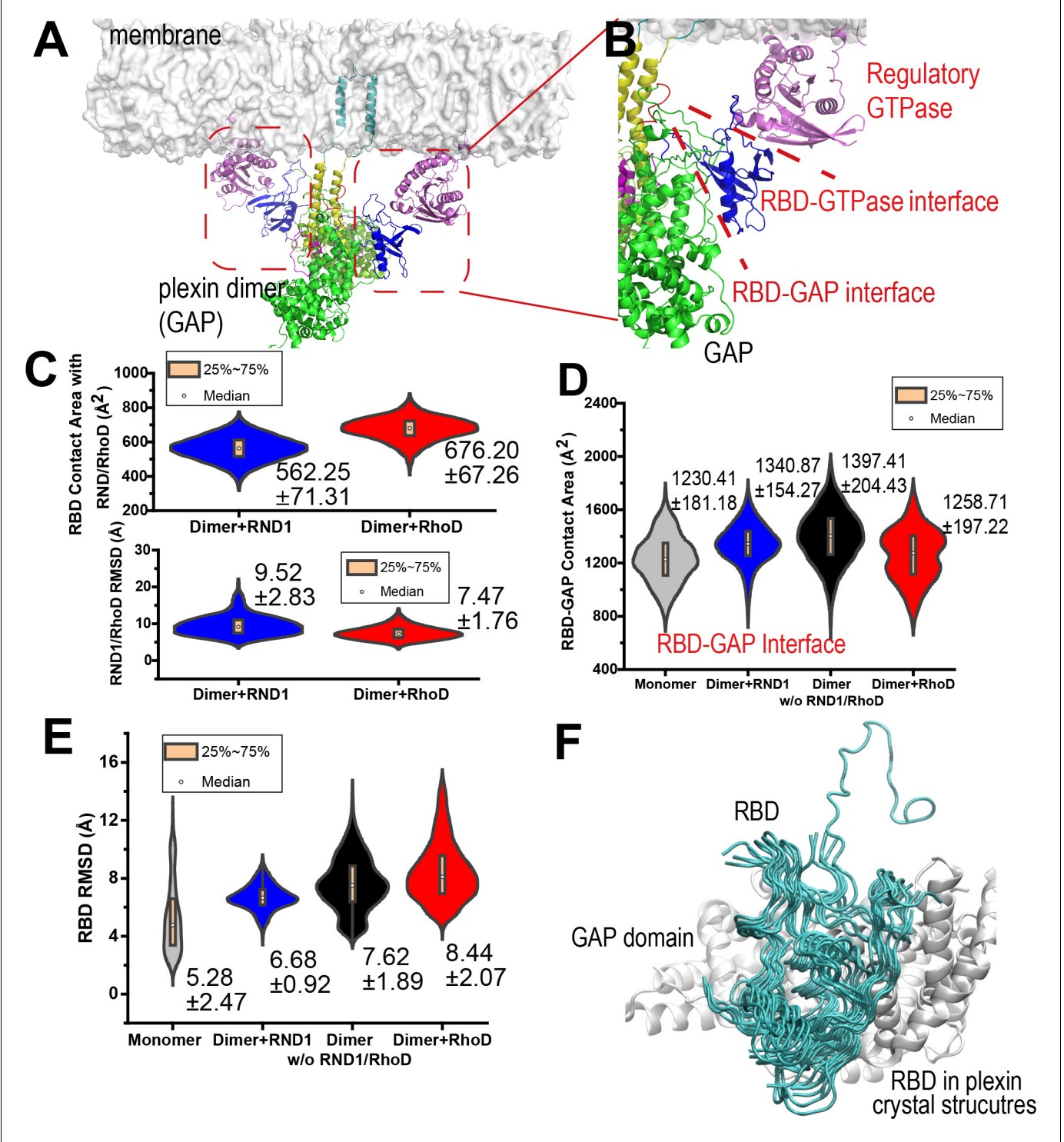

**Figure 4.** RND1- and RhoD-bound RBD domains are positioned differently with respect to their respective GAP domains. (**A**) RhoD-bound plexin dimer. (**B**) A close-up of a part of the plexin dimer illustrating the relative positions of the membrane, the RhoD (or RND1) GTPase (purple), the RBD domain (blue), the GAP domain (green), and the dimerization helices (yellow). (**C**) The RBD contact area of RND1 and RhoD, together with the relative flexibility of RND1 or RhoD relative to the respective RBD domains in terms of root mean square deviation (RMSD) of the catalytic domains with the RBD domains aligned. The RBD complexes of both RND1 and RhoD appear stable. The data sets were compiled from three 1-μs-long simulations each for RND1- and RhoD-bound plexin dimer; the individual distributions are shown in *Figure 3—figure supplement 1D*. As in similar panels, the average and RMSD for each distribution are shown. (**D**) The contact area of the RBD domains with their respective GAP domains. In addition to simulations of the RND1- and RhoD-bound plexin dimers, simulations of the plexin dimer and monomer with the RBD domains unoccupied are also included. RhoD-

*Figure 4 continued on next page*

*Figure 4 continued*

binding appears to moderately reduce the RBD-GAP contact area. The individual data sets from the simulations are shown in *Figure 3—figure supplement 1E*. (E) RBD flexibility relative to the GAP domain indicated by RMSD of the RBD domain measured with the GAP domain aligned. The data suggest that RND1 binding stabilizes the RBD conformation and RhoD binding destabilizes it. The data from simulations of the plexin monomer and dimer with unoccupied RBD suggests that the RBD domain is inherently flexible relative to the GAP domain. The individual data sets from the simulations are shown in *Figure 3—figure supplement 1F*. (F) Conformations of the RBD domain relative to the GAP domain in existing crystal structures of plexins. The GAP domain is aligned in generating this figure.

The online version of this article includes the following figure supplement(s) for figure 4:

**Figure supplement 1.** Additional data on root mean square deviation (RMSD) and domain-domain contact.

position while RND1 tends to stabilize RBD at that position. Since RBD binds stably with both RND1 and RhoD, the differential RBD positioning and dynamics may likely be attributable to the differential membrane interactions of RND1 and RhoD.

## RBD affects plexin dimerization via the buttress segment

The dimerization of plexins is mediated by their dimerization helices that are immediately C-terminal to the juxtamembrane helices (*Figure 1B*). The interaction between two dimerization helices in the dimer, which resembles coiled-coil interactions, is reinforced by Helix 11 of the GAP domain (*Figure 1C*; *Wang et al., 2013*). In crystal structures, Helix 11 is a stable helix, but the segment to its N-terminal is more variable structurally – it takes the form of a $3_{10}$ or an $\alpha$ helix in some crystal structures, but in many other structures it is disordered. When it is a $3_{10}$ or an $\alpha$ helix, it becomes an extension of Helix 11 and runs adjacent and in parallel to the dimerization helix, structurally reinforcing the interaction of the two dimerization helices in resemblance to a buttress. Based on this observation, we refer to it as the buttress segment (*Figure 1A, B*).

The RBD is connected to the plexin GAP domain by two linkers, a C-terminal and an N-terminal linker. The C-terminal linker (residues 1597–1662) is followed immediately by the buttress segment. This linker is long and partially disordered in crystal structures, especially in the part closer to the buttress segment. This suggests that this linker is conformationally highly flexible. The shorter N-terminal linker (residues 1482–1495) connects RBD to the bulk of the GAP domain and is packed against the buttress segment (*Figure 5A, B*). It is likely that the N and C linkers mediate the regulation of the buttress segment by the RBD since their conformations are expected to be closely coupled with the position of the RBD on one side and with the conformation of the buttress segment on the other.

Our simulations showed that the buttress interaction with the dimerization helix is minimal in a monomeric plexin, and this interaction increases substantially in plexin dimers (*Figure 5D* and *Figure 5—figure supplement 1A*). Importantly, with RhoD binding at the RBD, the buttress interaction with the dimerization helix in the plexin dimer is much reduced compared to that in the RND1-bound dimer or in the dimer where the RBD domains are unoccupied, suggesting that RhoD weakens the buttress interaction with the dimerization helix and potentially destabilizes the plexin dimer. In simulations of the RhoD-bound plexin dimer, the buttress segment lost its helical structure and gradually disengaged the dimerization helices (*Figure 5C*). In contrast, in simulations of the RND1-bound dimer both the helical structure and the interaction with the dimerization helices are much more stable (*Figure 5B*). The difference is reflected by the smaller contact area of the buttress segments and the dimerization helices in the RhoD-bound system than in the RND1-bound system (*Figure 5D* and *Figure 5—figure supplement 1A*). Moreover, the simulations showed that in the RhoD-bound dimer the pair of the dimerization helices was conformationally more variable than that in an RBD-unoccupied plexin dimer, and the dimerization helices in an RND1-bound dimer were less variable than the unoccupied dimer (*Figure 5E, F* and *Figure 5—figure supplement 1B*). This is consistent with the notion that RhoD binding destabilizes the plexin dimerization interface while RND1 binding may stabilize the dimer.

Based on these simulation results, we suggest that the differential membrane interaction of RND1 and RhoD propagates to the plexin dimerization interface and confers antithetic impact to plexin dimerization through the RBD domain and its N and C linkers (*Figure 5B*). RhoD binding destabilizes the RBD with respect to the GAP domain, destabilizing the buttress segment with

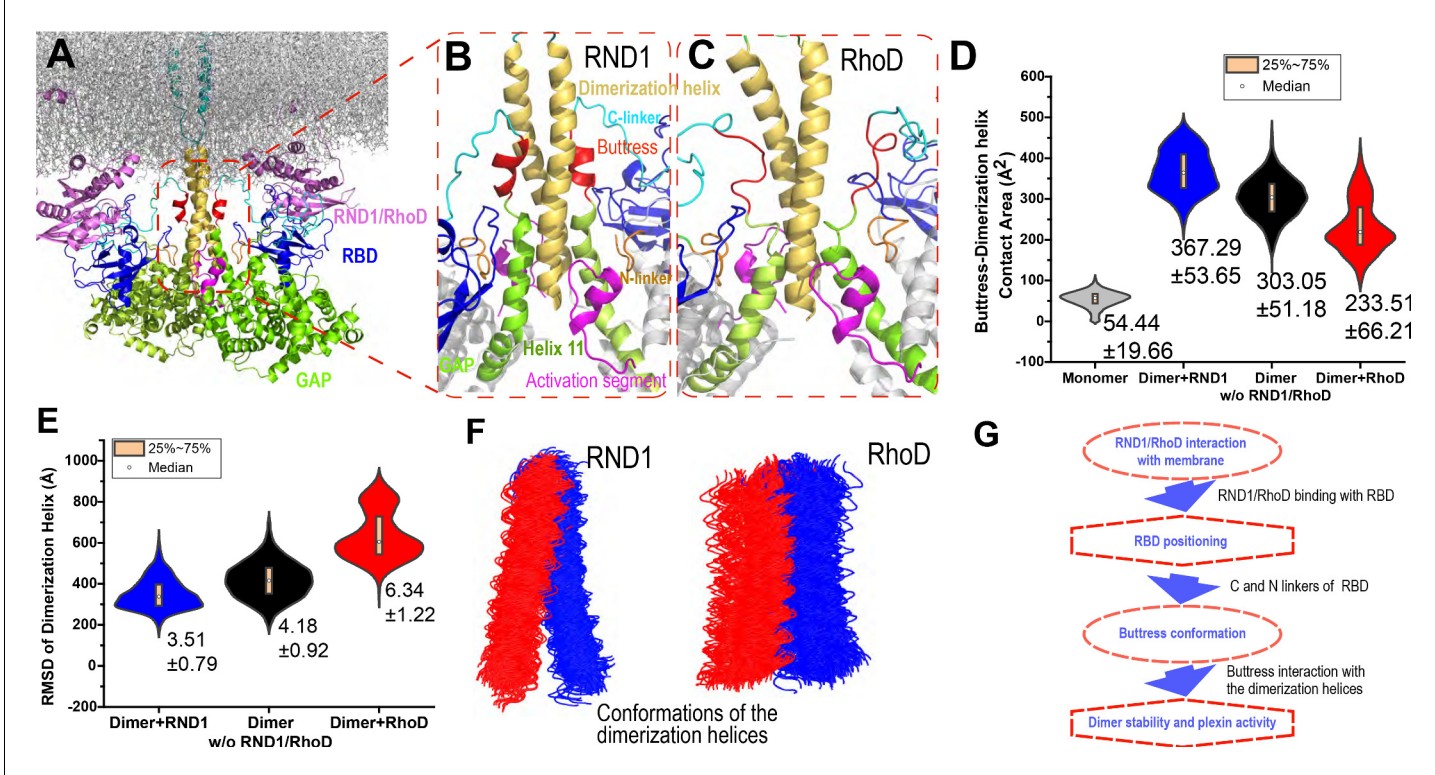

**Figure 5.** Interaction between the buttress segment and the dimerization helices. (**A**) RhoD- or RND1-bound plexin dimer. (**B**) Close-up of the RND1-bound dimer centered at the dimerization helices (yellow). The buttress segments (red), Helix 11 (green), the RBD domains (blue), the N (orange) and C (cyan) linkers of the RBD domains, and the activation segments (purple) are shown. (**C**) A similar close-up of the RhoD-bound dimer. (**D**) Contact area of the buttress segment with the dimerization helix. As shown, RND1 binding moderately raises the contact area, and RhoD reduces the contact area. The individual data sets from the simulations are shown in *Figure 3—figure supplement 1G*. (**E**) The root mean square deviation (RMSD) of the dimerization helices in plexin dimers as a measurement of the stability of the dimerization interface. RhoD binding clearly destabilizes the dimerization interface. The individual data sets from the simulations are shown in *Figure 3—figure supplement 1H*. The average and RMSD for each distribution are shown. (**F**) Snapshots of the dimerization helices. As shown, the dimerization helices are more flexible with respect to one another in the RhoD-bound plexin dimer. (**G**) A schematic summary of the mechanism by which RhoD and RND1 binding regulate plexin dimerization.
The online version of this article includes the following figure supplement(s) for figure 5:

**Figure supplement 1.** Additional data on root mean square deviation (RMSD) and the buttress-dimerization helices contact.

respect to the dimerization interface, and ultimately leads to destabilization of the dimer interface. In contrast, by the same RBD-centered route, RND1 binding helps stabilize the plexin dimer.

## Discussion

Plexins function in ways similar to a transistor in that they take two inputs and their responses to the primary input of semaphorin are regulated by the secondary input in the form of the Rho-family regulatory GTPase binding at the RBD domain. RND1 serves as a promoter of plexin signaling, while RhoD serves as an inhibitor. Our structural and molecular dynamics simulations and analyses suggest that the differential effects of RND1 and RhoD may arise from their differential interactions with the membrane. RND1 interacts with the membrane loosely and nonspecifically, while RhoD interacts with the membrane tightly with a specific interface. This difference gives rise to different positioning and dynamics of the RBD domain, which dictates the conformation of the buttress segment adjacent to the dimerization interface of plexin. We further showed that RhoD binding destabilizes the dimerization interface while RND1 binding helps stabilize the interface. In short, we propose an allosteric mechanism that regulates plexin dimerization involving cell membranes, the regulatory GTPases, the RBD domain, and the buttress segment (*Figure 5G*).

Our results on RND1 and RhoD offer a framework for the analysis of plexin regulation by Rho-family GTPases. We show that the antithetic roles of RND1 and RhoD result from two seemingly minor differences. First, RhoD furnishes a much shorter C-terminal tail than RND1, and consequently, RhoD is spatially more restrained to the membrane than RND1. Secondly, RhoD features a surface region that is rich in positively charged residue, which serves as the interface with membranes; these positively charged residues are not present in RND1. These two differences determine that RND1 and RhoD interact with the membrane differently and play different roles in plexin regulation. To experimentally validate or falsify this hypothesis, we suggest testing the effect of altering the C-terminal tails of RND1 and RhoD and the electrostatic properties of the putative membrane interface (*Figure 3G*). Specifically, mutating RND1 residues (e.g., Leu133, Glu138, Ser140, and Glu150) at the αI or α4 helices (the putative membrane-contacting region of RhoD) into positively charged arginines or lysines should impair the role of RND1 as a plexin activator, and conversely, mutating the lysines and arginines at these two helices of RhoD should impair the role of RhoD as a plexin inhibitor. By the same rationale, lengthening the C-terminal loop of RhoD should impair its inhibitory effect, while shortening the loop of RND1 should impair its activating effect. Likely, combinations of these two sets of modifications to RND1 and RhoD should confer a compound effect.

We analyzed the sequences of the Rho-family GTPases and, to our surprise, found that these two features are indeed correlated. The Rho GTPases with shorter C-terminal tails indeed tend to feature more positively charged residues at the putative membrane interface (*Figure 6*). This suggests that, besides RND1 and RhoD, other Rho-family GTPases may also be involved in regulations of plexin signaling, and that the Rho-family GTPases with short C-terminal tails may likely be downregulators and the other with long C-terminal tails likely upregulators. In cell biology, similar to plexin regulation by Rho-family GTPases, there are many other cases in which similar proteins in the same family interact with their target proteins almost identically yet achieve opposite regulatory effects. Simulations are an expedient platform to gain insight into such mechanisms.

In this study, we chose to focus on plexin A4 as a representative system in spite of the crystal structures of the intracellular domains being not available for A4. Unlike A4, for those plexins for which better structural data are available, direct functional data of regulation by Rho-family GTPases

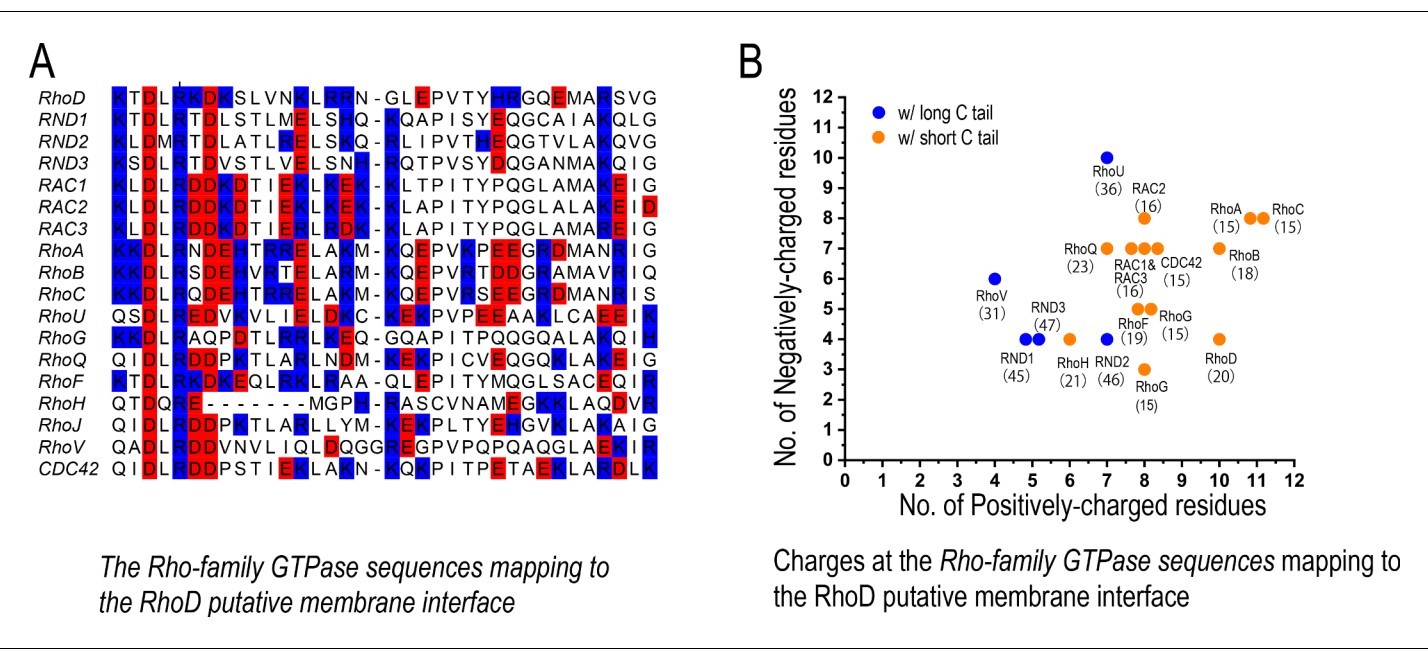

**The Rho-family GTPase sequences mapping to the RhoD putative membrane interface**

**Charges at the *Rho-family GTPase sequences* mapping to the RhoD putative membrane interface**

**Figure 6.** The charge distribution at the putative membrane interface of Rho-family GTPases. (**A**) Sequence alignment of Rho-family GTPases at the region of the putative membrane interface; red denotes negatively charged residues, and blue denotes positively charged residues. All members of the Rho family are included in this analysis with the exception of RHBT1, RHBT2, and RHBT3, which furnish another domain C-terminal to the catalytic domain. (**B**) The number of positively and negatively charged residues at the membrane interface. The protein name and the number of residues of its C-terminal tail are marked next to each data point herein. Long and short C tails are also color-coded.

are lacking. Plexin C1, which is arguably the best structurally characterized plexin in terms of the intracellular domains, is such an example. Even for C1, the structural information is incomplete as the structure of C1 complex with a Rho-family GTPase is not available. It is thus necessary to construct models from other plexin structures regardless of our choice of plexin system. We resorted to homology modeling (see Materials and methods) to construct A4 structures for simulations, considering the high level of sequence (35% or above overall) and structure similarly among the plexin family members, particularly in the dimer interface. The binding mode between class A plexins and Rho-family GTPases is particularly conserved, as shown by the numerous crystal structures, including that of plexin B2/RhoD presented in this paper. We therefore believe that the models of the plexin A4/RND1 and plexin A4/RhoD complexes are reliable. Moreover, our conclusion concerns mostly inter-domain interactions and thus is less likely to be sensitive to the structural details as the main driving force is the electrostatic interactions between the GTPases and membrane (*Figure 3G*), rather than any specific residue-residue interactions arising from a specific conformation. Reassuringly, our findings are supported by a recent study on plexin B1, which (*Li et al., 2021*) identified the functional importance of the buttress segment (or 'activation switch loop' as is referred therein) based on analysis of plexin enzymatic turnover, and showed that the segment helps stabilize the dimerization helix when the plexin active site is occupied by Rap.

Our results suggest that similar to many other signaling proteins, for plexin the membrane also plays an important role in its regulation. In a membrane environment of a high composition of negatively charged lipids such as POPS, PIP2, and PIP3, plexin signaling is likely more susceptible to negative regulation by RhoD. There are reports that plexin signaling activates the PI3K/AKT pathway, upon which PIP2 lipids in the membrane are phosphorylated and converted to more negatively charged PIP3 lipids (*Falkenburger et al., 2010*). Our findings raise the question as to whether down-regulation associated with RhoD binding may be a part of a negative feedback mechanism for plexin signaling involving the PI3K/AKT pathway.

## Materials and methods

### 1. Construction of the simulation systems

This research included eight simulation systems: RBD-RND1 and RBD-RhoD complexes, plexin monomer, plexin dimer with unoccupied RBD, two RND1-bound dimer systems, and two RhoD-bound dimer systems. Except for the RBD-RND1 and RBD-RhoD complexes, which are membrane-free, the other systems included the membrane and plexin transmembrane helix.

Lacking the crystal structure for plexin A4, we constructed one monomeric structure of the intracellular portion of plexin A4 using homology modeling. The sequence of mouse plexin A4 was taken from the NCBI website (http://www.ncbi.nlm.nih.gov/protein). The templates were selected according to the SWISS-MODEL searching results (http://swissmodel.expasy.org/) (*Bertoni et al., 2017*; *Guex et al., 2009*; *Waterhouse et al., 2018*), which were mainly the intracellular domain including mouse plexin A1 (PDB entry 3RYT), mouse plexin A3 (PDB entry 3IG3), mouse plexin B1 (PDB entry 3SU8), and human plexin C1 (PDB entry 4M8N), respectively. All the homology sequence identities of human plexin A4 with the mouse plexin A1, A3, B1, *h*plexin C1 were higher than 35%. The sequence alignment was done by T-coffee (*Lladós et al., 2018*), and the output alignment file was used to do homology modeling with Modeller 9.17 (*Webb and Sali, 2016*; *Fiser et al., 2000*). Modeller generated 100 structural models for the query sequence, and the one with the lowest estimated energy was selected for the construction of our simulation systems.

The plexin A4 dimer structure was obtained from superimposing the monomeric model of plexin A4 onto each protomer of the crystal structure of plexin C1 dimer (PDB entry 4M8N).

We also constructed the complex structure of RND1 with the RBD domain using Modeller 9.17. All the template structures selected in this research were downloaded from the Protein Data Bank (PDB) database. The templates for constructing RND1 we selected were the resolved crystal structures plexin A2 *h*RND1(PDB entry 3Q3J). The process of the Modeller-generating structures and the selection standard was the same as that used in constructing plexin structure. We separately resolved the RhoD-RBD complex structure using crystallography. A Palmitoyl group was covalently linked to Cys207 of the C-terminal tail of RhoD to produce a palmitoylated cysteine. This structure was incorporated into the simulation systems.

The CharmmGUI website (http://www.charmm-gui.org/) (*Jo et al., 2007*; *Jo et al., 2008*; *Wu et al., 2014*) was used to construct the systems containing the membrane. The membrane in the simulation systems comprised heterogeneous lipids. There were 1-palmitoyl-2-oleoyl-sn-glycero-3-phosphocholine (POPC) molecules in the upper leaflet and POPC and negatively charged POPS molecules with a ratio of 7:3 in the lower leaflet for the original dimer systems (*Arkhipov et al., 2013*; *van Meer et al., 2008*; *Zachowski, 1993*), and POPC and POPS and PIP2 molecules with a ratio of 70:25:5 in the lower leaflet in the control dimer systems, so there were negative charges in the inner membrane (*Jo et al., 2009*).

In order to ensure that the simulation results were not associated with the features of the initial models, we built two control simulation systems. We used the last snapshot of a 1-μs simulation of a RhoD-bound plexin dimer and swapped the RhoD molecules for RND1; similarly, we used the last snapshot of a 1-μs simulation of an RND1-bound plexin dimer and swapped the RND1 molecules for RhoD. 27 PIP2 molecules were added to the membrane, which constitute 5% of the lipids in the inner leaflet. We then performed three 0.5-μs-long simulations for each of these two systems.

The monomer system was a cubic box of $120 \times 120 \times 140$ Å$^3$ that contained 198,176 atoms in total, including water molecules and Na$^+$ and Cl$^-$ ions. The RBD-RND1 and RBD-RhoD systems take the form of a cubic box of $100 \times 100 \times 100$ Å$^3$ that contained 110,932 atoms and 104,447 atoms in total, respectively. The plexin dimer system with the RBD domains unoccupied was a cubic box of $190 \times 190 \times 170$ Å$^3$ that contained 639,794 atoms in total. The first RND1-bound plexin dimer system was a cubic box of $190 \times 190 \times 170$ Å$^3$ that contained 679,235 atoms in total. The first RhoD-bound plexin dimer system was a cubic box of $190 \times 190 \times 170$ Å$^3$ that contained 679,086 atoms in total. The control RND1-bound plexin dimer system was a cubic box of $180 \times 180 \times 170$ Å$^3$ that contained 580,115 atoms in total. The control RhoD-bound plexin dimer system was a cubic box of $180 \times 180 \times 190$ Å$^3$ that contained 648,627 atoms in total. The dimensions of the simulation boxes were chosen so that the minimum distance of any protein in a system was greater than 10 Å to the edge. Na$^+$ and Cl$^-$ ions were added to maintain physiological salinity (150 mM) and to obtain a neutral charge for the system. All the components in the system including POPC, POPS, PIP2, and protein, as well as palmitoylated lipid, were parameterized using the CHARMM36 force field (*Lee et al., 2016*) and TIP3P water model (*Jorgensen et al., 1983*).

The above dimer system with the RBD domains unoccupied system was also used to set up the G-protein-bound dimer system. When the system of two plexin monomer inserted into the membrane was generated, the initial placement of the RND1 molecule bound to RBD of plexin was determined by first superimposing one monomer structure in the dimer on the complex of RBD bound with RND1 (PDB entry 3Q3J) with the RBD domains aligned, followed by superimposing one RND1 structure on the RND1 in the complex structure encoded 3Q3J. The RND1 structure was placed at the targeting position. Meanwhile, both the GTP molecule and magnesium (Mg$^{2+}$) ion in the complex (PDB entry 3Q3J) were superimposed on the RND1 structure. For the other monomer, the RND1 structure as well as the GTP molecule and Mg$^{2+}$ were also placed at the corresponding positions in the same way. Finally, the whole system of the RND1-bound plexin dimer inserted into the membrane was setup. The RND1-bound dimer system was placed in a cubic box of $190 \times 190 \times 170$ Å$^3$ and 679,235 atoms in total in the system. The RhoD-bound dimer system was a cubic box of $190 \times 190 \times 170$ Å$^3$ that contained 679,086 atoms in total. Both RhoD and RND1 were GTP-bound in the systems with Mg$^{2+}$ coordinating the GTP binding.

## 2. MD simulations

Each initial simulation system was equilibrated under NPT ensemble at 1 bar and 300 K for 5 ns, after energy minimization (50,000 steps) and a preliminary NVT equilibration (500 ps) with the position restraint applied on the heavy atoms of the protein with a force constant of 10 kJ/mol/Å$^2$. Periodic boundary condition (PBC) was imposed on the system to eliminate the boundary effect. A cutoff distance of 12 Å was set for van der Waals interactions, and the long-range electrostatic interactions were treated by the Particle Mesh Ewald (PME) method (*Darden et al., 1993*). LINCS algorithm (*Hess et al., 1997*) was used to constrain the covalent bonds involving hydrogen atoms. The time step was set to 2.5 fs. The temperature was controlled by the Langevin thermostat with a collision frequency of 2.0 ps$^{-1}$, and the Berendsen barostat (*Berendsen et al., 1984*) was used to control the pressure at 1.0 atm. All MD simulations were performed using Gromacs 5.1.3 on Tianhe Supercomputer. Each of the simulations of the monomer and the dimer system with unoccupied RBD

domains was 0.5-μs long, and each of the simulations of the RND1- and RhoD-bound dimer was 1-μs long.

## 3. Trajectories analysis

### 3.1 Protein-protein contact area calculation

All the protein-protein contact areas were calculated using Gromacs command 'gmx sasa'.

### 3.2 RMSD analysis

RMSD of an RND1 or RhoD as an indicator of its position relative to the plexin

The RMSD calculation was carried out by first aligning the system by the Cα atoms of the GAP domain of the plexin protomer to which the GTPases is bound to, and then the RMSD was calculated using the Cα atoms of the GTPase with respect to their initial positions in the aligned simulation system.

RMSD of RND1 or RhoD as an indicator of its position relative to the RBD domain

The RMSD calculation was carried out by first aligning the system by the Cα atoms of the RBD domain of the plexin protomer to which the GTPases is bound to, and then the RMSD was calculated using the Cα atoms of the GTPase with respect to their initial positions in the aligned simulation system.

RMSD of RBD as an indicator of its position relative to the corresponding GAP domain

The RMSD calculation was carried out by first aligning the system by the Cα atoms of the GAP domain of the same plexin protomer, and then the RMSD was calculated using the Cα atoms of the RBD with respect to their initial positions in the aligned simulation system.

RMSD of the dimerization helix as an indicator of the stability of the dimer interface

The dimerization helices were first aligned using their Cα atoms, and then the RMSD was calculated using the Cα atoms with respect to their initial position.

### 3.3 The metric for protein-membrane interaction

For each residue of the protein in each simulation snapshot, the number of any lipid molecules within 5 Å of any atom of the residue is calculated. This number was averaged over each simulation (with the first 0.3 μs of the simulation ignored) for each protein residue as a metric for the residue's membrane interaction.

## 4. Sequence alignment

The sequences of human RND1 and RhoD were downloaded from the NCBI website. The sequence alignment of the GTPases of the Rho family was performed using the UniProt website (https://www.uniprot.org/).

## 5. Protein expression and purification

The coding region of the intracellular region of mouse plexin B2 with the juxtamembrane region removed (residues 1274–1842) was cloned into a modified pET-28(a) vector (Novagen) that encodes an N-terminal His6-tag followed by a recognition site for human rhinovirus 3C protease. The plasmid was transformed into the *Escherichia coli* strain ArcticExpress (DE3) (Stratagene). ArcticExpress (DE3) carrying the expression plasmid was cultured at 37℃ in 100–120 mL LB medium in the presence of gentamycin overnight. Bacterial cells were scaled up at 30℃ to reach OD600 2.0 in TB medium. Protein expression was induced by 0.2 mM IPTG at 10℃ overnight. Cells were harvested by centrifugation and resuspended in Buffer A containing 10 mM Tris (pH 8.0), 500 mM NaCl, 5% glycerol (v/v), 20 mM imidazole, and 3 mM β-mercaptoethanol. Cells were lysed with a Avestin C3 disruptor (Avestin) and subjected to centrifugation. The plexin protein in the supernatant was

captured using a 1 mL HisTrap FF column (GE Healthcare) and eluted by Buffer B containing 10 mM Tris (pH 8.0), 500 mM NaCl, 5% glycerol (v/v), 250 mM imidazole, and 3 mM β-mercaptoethanol. The protein was treated with recombinant human rhinovirus 3C protease at 4°C overnight to remove the N-terminal His6-tag. The tag-removed protein was loaded to a Resource Q anion-exchange column (GE Healthcare) and eluted with a linear NaCl gradient (10 mM to 300 mM). Fractions containing plexin B2 were pooled and subjected to size exclusion chromatography with a Superdex 200 GL 10/30 column (GE Healthcare) equilibrated with Buffer C containing 20 mM Tris (pH 8.0), 150 mM NaCl, 10% glycerol (v/v), and 2 mM DTT. Purified proteins were concentrated and stored at −80°C.

The coding region of human RhoD (residues 8–194) with the Q75L mutation, which renders the protein catalytically dead and therefore does not hydrolyze GTP, was cloned into the above-mentioned modified pET-28(a) vector. The plasmid was transformed into the bacterial strain BL21 (DE3). Protein expression was induced by 0.2 mM IPTG at 16°C overnight. The protein purification procedure was similar to that for plexin B2, except that all the buffers contained 2 mM $MgCl_2$. The RhoD protein with the Hist6-tag removed was subjected to the final purification step with a Superdex 75 GL 10/30 column with Buffer D containing 20 mM Tris (pH 8.0), 250 mM NaCl, 10% glycerol (v/v), 2 mM $MgCl_2$, and 2 mM DTT.

To load the protein with GMP-PNP (guanosine 5′-[β,γ-imido]triphosphate) for crystallization, the purified RhoD protein was incubated with GMP-PNP at 20-fold molar ratio to the protein in the exchange buffer containing 20 mM Tris (pH 8.0), 250 mM NaCl, 5% glycerol (v/v), 7.5 mM EDTA, and 1 mM DTT at RT for 2 hr. After the incubation, 20 mM $MgCl_2$ was added to stop the exchange reaction. The protein was then subjected to gel filtration chromatography on a Superdex 75 GL 10/30 column equilibrated with Buffer D to remove excess GMP-PNP.

## 6. Crystallization, X-ray data collection, and structure determination

Plexin B2 and GMP-PNP-loaded RhoD were mixed at 1:1 molar ratio in a buffer containing 10 mM Tris (pH 8.0), 150 mM NaCl, 10% glycerol (v/v), 2 mM $MgCl_2$, 2 mM TCEP, and 100 µM GMP-PNP to form the complex. The total protein concentration of the complex for crystallization was 6 mg/mL. The complex was crystallized initially at 20°C in 0.2 M $MgCl_2$ and 20% PEG3350 (w/v) in sitting-drop 96-well plates. Crystals large enough for data collection were grown for over a month with sitting-drop or hanging-drop vapor diffusion at 20°C in 0.2 M $MgCl_2$, 22% PEG3350 (w/v), and 100 mM MIB (pH 6.8, sodium malonate, imidazole, and boric acid mixed at 2:3:3 molar ratio). Crystals were cryo-protected using the crystallization buffer supplemented with 25% glycerol and flash cooled in liquid nitrogen. Diffraction data were collected at 100 K at the beamline 19ID at the advance photon source (Argonne, IL). Data were indexed, reduced, and scaled with the software HKL2000 (*Otwinowski and Minor, 1997*). Molecular replacement using RND1 (PDB ID: 2REX) as the search model with the program phaser (*McCoy et al., 2007*) found two copies of RhoD in the asymmetric unit. However, repeated search using various full-length intracellular region of plexin models failed to yield any solution. In the end, the RBD of plexin B1 (PDB ID: 2REX) as the search model led to the solution of two copies of plexin B2-RBD in the asymmetric unit. It is likely that the full-length intracellular region of plexin B2 was degraded during the prolonged incubation at 20°C in the crystallization drops, which separated the RBD from the rest of the protein. The RBD formed the complex with RhoD, which crystallized at the end. The initial model from molecular replacement was manually modified in Coot (*Emsley et al., 2010*) and refined using Phenix (*Liebschner et al., 2019*). The density clearly showed that the two RBD domains formed a domain-swapped dimer, with the swap occurring between residues 1509 and 1510. As a result, the N-terminal segment (residues 1463–1509) from the first molecule and the C-terminal segment (residues 1510–1565) from the second molecule pack together to form one RBD, and vice versa. The conformation of the RBD formed in this manner is very similar to other RBD structures in the database, and its binding mode with RhoD is very similar to that in other RhoGTPase/RBD complexes. This domain-swapped dimer cannot form in the context of the intact plexin, and therefore is unlikely to have any biological significance. The refined structure was validated by using Molprobity as implemented in Phenix (*Williams et al., 2018*). The data collection and structure refinement statistics are summarized in *Supplementary file 1.*

## Acknowledgements

YL was supported by the National Natural Science Foundation of China (21806004) and Boya Post-doctoral Fellowship at Peking University. PK was supported by China NSAF Grant U1430237. XZ was supported in part by grants from the National institutes of Health (R35GM130289) and the Welch Foundation (I-1702). XZ is a Virginia Murchison Linthicum Scholar in Medical Research at UTSW. Results shown in this report are derived from work performed at the Argonne National Laboratory, Structural Biology Center at the Advance Photon Source. Argonne is operated by University of Chicago, Argonne for the US Department of Energy, Office of Biological and Environmental Research under contract DE-AC02-06CH11357. CS was supported by the National Natural Science Foundation of China (21873006 and 32071251), and the Ministry of Science and Technology of China (2016YFA0500401). The MD simulations were performed on TianHe-1A at the National Supercomputer Center in Tianjin, China.

## Additional information

### Competing interests

Yibing Shan: Reviewing Editor, *eLife*. The other authors declare that no competing interests exist.

### Funding

| Funder | Grant reference number | Author |
|---|---|---|
| National Natural Science Foundation of China | 21806004 | Yanyan Liu |
| NSAF Joint Fund | U1430237 | Pu Ke |
| National Institutes of Health | R35GM130289 | Xuewu Zhang |
| Welch Foundation | I-1702 | Xuewu Zhang |
| National Natural Science Foundation of China | 21873006 | Chen Song |
| National Natural Science Foundation of China | 32071251 | Chen Song |
| Chinese Ministry of Science and Technology | 2016YFA0500401 | Chen Song |

The funders had no role in study design, data collection and interpretation, or the decision to submit the work for publication.

### Author contributions

Yanyan Liu, Data curation, Formal analysis, Investigation, Visualization, Writing - original draft, Writing - review and editing; Pu Ke, Data curation, Formal analysis; Yi-Chun Kuo, Yuxiao Wang, Data curation, Formal analysis, Visualization; Xuewu Zhang, Conceptualization, Funding acquisition, Writing - original draft, Writing - review and editing; Chen Song, Supervision, Funding acquisition, Methodology, Writing - original draft, Writing - review and editing; Yibing Shan, Conceptualization, Supervision, Funding acquisition, Investigation, Visualization, Writing - original draft, Writing - review and editing

### Author ORCIDs

Xuewu Zhang https://orcid.org/0000-0002-3634-6711
Chen Song http://orcid.org/0000-0001-9730-3216
Yibing Shan https://orcid.org/0000-0002-3865-8110

### Decision letter and Author response

Decision letter https://doi.org/10.7554/eLife.64304.sa1
Author response https://doi.org/10.7554/eLife.64304.sa2

## Additional files

### Supplementary files
- Supplementary file 1. Diffraction data and refinement statistics of the crystal structure.
- Transparent reporting form

### Data availability
Diffraction data have been deposited in PDB under the accession code 7KDC. Simulation data have been deposited in ZONODO database.

The following datasets were generated:

| Author(s) | Year | Dataset title | Dataset URL | Database and Identifier |
|---|---|---|---|---|
| Kuo C, Wang Y, Zhang X | 2021 | Crystal structure of the complex between RhoD and the plexin B2-RBD domain | https://www.rcsb.org/structure/7KDC | RCSB Protein Data Bank, 7KDC |
| Liu Y, Ke P, Kuo Y-C, Wang Y, Zhang X, Song C, Shan Y | 2021 | A putative structural models underlying the antithetic effect of homologous RND1 and RhoD GTPases in plexin regulation | https://zenodo.org/record/4751081#.YKw1zS18jfY | zenodo, 4751081#.YKw1zS18jfY |

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
