## [Decision Letter]

**Acceptance summary:**

Using molecular dynamics simulations, along with some structural and bioinformatics analyses, the authors of this manuscript try to explain why two closely related GTPase homologs, RND1, and RhoD, have antithetic effects on plexin regulation. Because plexin signaling regulates several critical biological processes and several diseases are associated with disrupted plexin signaling, understanding the basis of plexin inhibition and activation could provide key molecular insights into this plexin-mediated pathway; as a result, this work will have a broad audience. The model put forward in this work is a valuable framework for generating additional structure-based hypotheses aimed at teasing out further insights into the antithetic effects of RND1 and RhoD on plexin regulation.

**Decision letter after peer review:**

Thank you for submitting your article "The structural mechanism underlying the antithetic effect of homologous RND1 and RhoD GTPases in plexin regulation" for consideration by *eLife*. Your article has been reviewed by 3 peer reviewers, one of whom is a member of our Board of Reviewing Editors, and the evaluation has been overseen by José Faraldo-Gómez as the Senior Editor. The following individuals involved in review of your submission have agreed to reveal their identity: Alex Dickson (Reviewer #2); Matthieu Chavent (Reviewer #3).

The reviewers have discussed the reviews with one another and the Reviewing Editor has drafted this decision to help you prepare a revised submission.

Summary:

Using molecular dynamics simulations, along with some structural and bioinformatics analyses, the authors of this manuscript try to explain why two closed related GTPase homologs, RND1, and RhoD, have antithetic effects on plexin regulation. Because plexin signaling regulates several critical biological processes and several diseases are associated with disrupted plexin signaling, understanding the basis of plexin inhibition and activation could provide key molecular insights into this plexin-mediated pathway; as a result, this work will have a broad audience. However, the reviewers have some reservations that the manuscript's strong claims are based on limited simulation data of a large multi-component molecular system whose structure was built using homology modeling. Therefore, aspects of the structure determination and modeling need to be clarified to fully support the authors' claims. The reviews have also suggested several improvements that could make the paper more readable and make the statistical analysis of the molecular dynamics simulations more rigorous.

The reviews' consensus was that the work presented here is potentially suitable for publication; the combination of X-ray crystallography with extensive modeling appears to provide new insights into the molecular basis for the interaction of plexin with each GTPase. However, as the structural effects described are quite subtle, additional analyses and modeling will be necessary to fully validate the mode of association of these two GTPases with both the membrane and the plexin dimer.

We request that the authors make a note of and respond to the following comments, in addition to the essential revisions listed below:

Essential revisions:

1. The primary concern of the reviews is the quality and built-in uncertainties of the initial models. Currently, the manuscript lacks consideration and discussion of the sensitivity of the results presented to the initial models. As the authors are most likely aware, the effect of the initial structure on the observations made on an MD trajectory can extend beyond the trajectory itself, depending on its length and the type of observable considered. Based on the data provided, it is unclear whether the conclusions are insensitive to the assumptions made in the construction of the initial homolog model. As such, the reviewers request that the authors carry out additional simulations with alternative models that are similarly plausible and yet meaningfully different from the models used in current version of the manuscript.

2. On a related note, the authors should comment and include a discussion in the updated manuscript on the validity of the domain-swapped X-ray structure.

3. The simulations are short relative to the state of the art, which is especially important for such large systems. To achieve full convergence, longer simulations or enhanced sampling is techniques may be required. Is there a reason why enhanced sampling methods were not employed? What metrics were employed to ascertain the statistical significance of the results presented.

4. The conclusions are declarative, but simulation results can only make predictions, and they should be stated as such. For instance, the authors state in the last sentence of the first paragraph in the Discussion that: "In short, we reveal an allosteric mechanism that regulates plexin dimerization involving cell membranes, the regulatory GTPases, the RBD domain, and the buttress segment (Figure 5F)." At best, one can say that "…we reveal a possible allosteric mechanism." Even if the additional modeling requested above demonstrates that the conclusions are robust to the initial models, the authors will need to soften their claims and update the title accordingly to reflect that these are just predictions. Otherwise, their simulation data on its own are not strong enough to support their claims.

5. As mentioned by the authors in the Discussion section, the inner leaflet of the membrane is constituted by different negatively charges lipids which are known to have a role in signaling. One can cite especially the PIP2/3 lipids. Here, the authors have used a membrane composed on 7:3 ratio of POPC:POPS. It would be useful for the reader to explain this choice and maybe to run new simulations to see the action of the PIP2/3 lipids on the plexin/GTPases complex. It would also be valuable for the reader to see if negatively charged lipids may be differently attracted by RhoD and RND1. This may reinforce the authors' hypothesis and also inform the readership on how protein may drive the formation of lipid nanoclusters, which may have consequences for GTPases signaling.

6. The system models were constructed with a membrane composed of POPC in the outer leaflet and of a ratio of 7:3 POPC:POPS for the inner leaflet. While POPS lipid can be seen as a proxy for negatively charged lipids, there are quite important negatively charged lipids missing, such as PIP2 and PIP3. It is now quite clear that these lipids can play a role in cell signaling. Thus, adding PIP2/3 lipids into the model may further validate the authors' claims with a more biologically relevant membrane.

7. The author claims that the movement of RhoD alpha helix αi is due to allosteric changes. Displaying the full unit cell shows crystal packing contacts, which may affect the position of this αi helix. Atomistic simulations may help to assess the stability of the structure of RhoD-RBD complex in solution and confirm the position of the αi helix.

8. It is unclear how the authors have chosen the orientation of RND1 and RhoD towards the membrane. Are there specific references mentioning the position of RND1 and RhoD – or other homologous GTPases – towards the membrane? Would it be possible to randomly position these structures away from the membrane and perform MD simulations (maybe using low-resolution representations such as CG models to save computing time) to assess the preferred positioning of the respective structures?

9. Given the manuscript's bold claim, the authors must include a discussion about the testable hypotheses the emerge from their work and how they can be tested.

10. A key point of this study is that what differentiates RND1 and RhoD are the lengths and the number of positively charged residue in the C-terminal tail, all of which the author could obtain from the bioinformatic analysis as presented in Figure 6, without requiring simulations. Could the reviewers comment on the value added by the simulations?

11. The structures of the different models considered should be made available on a citable website such as ZENODO. This would be useful to other research teams (both computational and experimental) to continue to develop new hypotheses from this work and continue to build further experiments. This will be beneficial both for the modeling community (to expand on this work) and the authors (to be credited beyond the results presented in this manuscript).

---

## [Author Response]

Essential revisions:1. The primary concern of the reviews is the quality and built-in uncertainties of the initial models. Currently, the manuscript lacks consideration and discussion of the sensitivity of the results presented to the initial models. As the authors are most likely aware, the effect of the initial structure on the observations made on an MD trajectory can extend beyond the trajectory itself, depending on its length and the type of observable considered. Based on the data provided, it is unclear whether the conclusions are insensitive to the assumptions made in the construction of the initial homolog model. As such, the reviewers request that the authors carry out additional simulations with alternative models that are similarly plausible and yet meaningfully different from the models used in current version of the manuscript.

We should clarify that the construction of the initial models entailed little discretionary decision by the modeller. The construction was largely determined by existing crystal structures of plexin dimer and crystal structures of plexin RBD domain bound with Rho small GTPases. The plexin dimer was so placed relative to the membrane such that the two plexin molecules are symmetric in terms of their geometric positions to the membrane. We have added text to make this point explicit.

That said, we agree that additional simulations will be helpful to solidify our case. The key simulation result is that, starting from the initial models, our simulations generated RND1- and RhoD-bound plexin models in which RND1 interacts loosely while RhoD interacts extensively with the membrane. For the additional control simulations, we swapped RND1 and RhoD and produced new initial models that contradict the simulation results. In these new initial models RND1 interacts with the membrane extensively while RhoD interacts with the membrane loosely. We launched three additional simulations for each of the new models. Reassuringly, we observed that the previous simulation result was confirmed despite the new initial models. Steadily, the RND1 membrane interaction decreased and the RhoD membrane interaction increased and surpassed RND1 in the simulations. This is now reported in the revised manuscript (Figure 3I, Figure 3-supplement 1B and 1C). We hope these new simulations sufficiently addressed the reviewers’ concern that the simulation results may be dependent on the initial models.

2. On a related note, the authors should comment and include a discussion in the updated manuscript on the validity of the domain-swapped X-ray structure.

The domain-swapped dimer of the RBD in the X-ray structure is likely a crystallographic artifact, which happens relatively often in protein crystallization. We therefore do not assign any biological significance to it. Instead, we show in the manuscript the reconstructed un-swapped RBD monomer, which is very similar to previously published structures of the RBD from other plexin family members.

3. The simulations are short relative to the state of the art, which is especially important for such large systems. To achieve full convergence, longer simulations or enhanced sampling is techniques may be required. Is there a reason why enhanced sampling methods were not employed? What metrics were employed to ascertain the statistical significance of the results presented.

These simulation systems of plexin dimers are relatively large, each includes ~640,000 atoms, which constrained the lengths of our simulations. We were interested in applying enhanced sampling methods such as replica exchange. In fact, our original plan was to run plain MD simulations and gain insight into plexin conformational dynamics, before we determine an appropriate enhance sampling methods. It was fortunate and fortuitous that the plain MD simulations were enough to provide relatively robust data that led to a meaningful result. In retrospect, we believe this is because the convergence of the protein-membrane interaction is relatively fast compared to large protein conformational changes.

4. The conclusions are declarative, but simulation results can only make predictions, and they should be stated as such. For instance, the authors state in the last sentence of the first paragraph in the Discussion that: "In short, we reveal an allosteric mechanism that regulates plexin dimerization involving cell membranes, the regulatory GTPases, the RBD domain, and the buttress segment (Figure 5F)." At best, one can say that "…we reveal a possible allosteric mechanism." Even if the additional modeling requested above demonstrates that the conclusions are robust to the initial models, the authors will need to soften their claims and update the title accordingly to reflect that these are just predictions. Otherwise, their simulation data on its own are not strong enough to support their claims.

This point is well taken. We now consistently recalibrated our statements. For example, the mentioned statement now reads “In short, we *propose* an allosteric mechanism that regulates plexin dimerization involving cell membranes, the regulatory GTPases, the RBD domain, and the buttress segment (Figure 5G).” We also revised the title to be “*A putative* structural mechanism underlying the antithetic effect of homologous RND1 and RhoD GTPases in plexin regulation”.

5. As mentioned by the authors in the Discussion section, the inner leaflet of the membrane is constituted by different negatively charges lipids which are known to have a role in signaling. One can cite especially the PIP2/3 lipids. Here, the authors have used a membrane composed on 7:3 ratio of POPC:POPS. It would be useful for the reader to explain this choice and maybe to run new simulations to see the action of the PIP2/3 lipids on the plexin/GTPases complex. It would also be valuable for the reader to see if negatively charged lipids may be differently attracted by RhoD and RND1. This may reinforce the authors' hypothesis and also inform the readership on how protein may drive the formation of lipid nanoclusters, which may have consequences for GTPases signaling.

The 7:3 ratio of POPC:POPS for the inner membrane leaflet was chosen to mimic the abundance of anionic lipids in the mammalian plasma membrane (Zachowski, 1993; van Meer et al., 2008). This is to mimic ~20% of POPS and ~10% of PIP, PIP2, and PIP3 combined. This is now clarified in the manuscript. Following the reviewers’ suggestion, in the new simulations for this revision, in the inner leaflet we included 25% POPS and 5% PIP2. Our analysis showed that the charged lipids tend to concentrate at the membrane-RhoD interface (Figure 3-supplement 1E and 1F). This is consistent with the reviewer’s notion that the membrane environment may have consequences for Rho GTPase-mediated plexin regulation.

6. The system models were constructed with a membrane composed of POPC in the outer leaflet and of a ratio of 7:3 POPC:POPS for the inner leaflet. While POPS lipid can be seen as a proxy for negatively charged lipids, there are quite important negatively charged lipids missing, such as PIP2 and PIP3. It is now quite clear that these lipids can play a role in cell signaling. Thus, adding PIP2/3 lipids into the model may further validate the authors' claims with a more biologically relevant membrane.

We have now included PIP2 lipids in the new simulations. Please see the reply to the previous comments.

7. The author claims that the movement of RhoD alpha helix αi is due to allosteric changes. Displaying the full unit cell shows crystal packing contacts, which may affect the position of this αi helix. Atomistic simulations may help to assess the stability of the structure of RhoD-RBD complex in solution and confirm the position of the αi helix.

Following this comment, we have now simulated RBD/RhoD and RBD/RND1 complexes in solution. The simulations showed that both RhoD and RND1 bind with the RBD stably, but RND1 conformation is slightly more flexible than RhoD in the complex, both in terms of the entire GTPase domain and in terms of the alpha-I helix (Figure 2C, 2D, and 2E). We suggest that the relative stability of the alpha-I helix of RhoD is consistent with the notion that in plexin regulation RhoD is engaged in stable membrane interaction involving this helix. This is reported in the revised manuscript.

8. It is unclear how the authors have chosen the orientation of RND1 and RhoD towards the membrane. Are there specific references mentioning the position of RND1 and RhoD – or other homologous GTPases – towards the membrane? Would it be possible to randomly position these structures away from the membrane and perform MD simulations (maybe using low-resolution representations such as CG models to save computing time) to assess the preferred positioning of the respective structures?

Please also refer to our reply to Comment 1 of this review. We believe there is a misunderstanding here as to how our models were constructed. We have made revisions to clarify this point. To be clear, the positioning of RND1 or RhoD with respect to the membrane was entirely determined by existing crystal structures and simple symmetry. There is no room for manipulation by the modeler. The pose of RND1 or RhoD with the RBD domain is defined by crystal structures, and the RBD poses with respect to the plexin dimer as a whole is defined by crystal structures of plexin dimers in which the RBD domains are resolved. Thus the position of RND1 or RhoD in the plexin dimer is defined by crystal structures. We placed the plexin dimer relative to the membrane by symmetry, so that the two plexin protomers and the two RND1 or RhoD molecules they respectively bind are positioned identically relative to the membrane. In simulation RBD binding with RND1 or RhoD was largely stable, and RBD moved with RND1 and RhoD essentially as a whole.

9. Given the manuscript's bold claim, the authors must include a discussion about the testable hypotheses the emerge from their work and how they can be tested.

We thank the reviewers for this suggestion. We have now included some discussion as how what specific mutagenesis experiments should be considered to validate the hypothesis. In discussions we wrote that “To experimentally validate or falsify this hypothesis, we suggest testing the effect of altering the C-terminal tails of RND1 and RhoD and the electrostatic properties of the putative membrane interface (Figure 3G). […] By the same rationale, lengthening the C-terminal loop of RhoD should impair its inhibitory effect, while shortening the loop of RND1 should impair its activating effect. Likely, combinations of these two sets of modifications to RND1 and RhoD should confer a compound effect.”

10. A key point of this study is that what differentiates RND1 and RhoD are the lengths and the number of positively charged residue in the C-terminal tail, all of which the author could obtain from the bioinformatic analysis as presented in Figure 6, without requiring simulations. Could the reviewers comment on the value added by the simulations?

We think the labeling of our figure might have misled the reviewer. We have revised the figure (Figure 6). Instead of plotting the number of charged residues in the C tail, Figure 6B actually plotted the number of charged residues of the putative membrane interaction interfaces of RND1 and RhoD, which were inferred from the simulations. Without simulation, there would be no knowing of the putative membrane interface.

11. The structures of the different models considered should be made available on a citable website such as ZENODO. This would be useful to other research teams (both computational and experimental) to continue to develop new hypotheses from this work and continue to build further experiments. This will be beneficial both for the modeling community (to expand on this work) and the authors (to be credited beyond the results presented in this manuscript).

We agree with the reviewer. We have deposited our models into ZENODO.